# Effect of luminal surface structure of decellularized aorta on thrombus formation and cell behavior

**Mako Kobayashi**[1‡], **Masako Ohara**[2‡], **Yoshihide Hashimoto**[1], **Naoko Nakamura**[2], **Toshiya Fujisato**[3], **Tsuyoshi Kimura**[1], **Akio Kishida**[1] *

**1** Institute of Biomaterials and Bioengineering, Tokyo Medical and Dental University, Chiyoda-ku, Tokyo, Japan, **2** Department of Bioscience and Engineering, Shibaura Institute of Technology, Minuma-ku, Saitama-shi, Saitama, Japan, **3** Department of Biomedical Engineering, Osaka Institute of Technology, Asahi-ku, Osaka, Japan

‡ These authors share first authorship on this work.
* kishida.mbme@tmd.ac.jp

**Data Availability Statement:** All relevant data are within the manuscript and its Supporting Information files.

## Abstract

Due to an increasing number of cardiovascular diseases, artificial heart valves and blood vessels have been developed. Although cardiovascular applications using decellularized tissue have been studied, the mechanisms of their functionality remain unknown. To determine the important factors for preparing decellularized cardiovascular prostheses that show good *in vivo* performance, the effects of the luminal surface structure of the decellularized aorta on thrombus formation and cell behavior were investigated. Various luminal surface structures of a decellularized aorta were prepared by heating, drying, and peeling. The luminal surface structure and collagen denaturation were evaluated by immunohistological staining, collagen hybridizing peptide (CHP) staining, and scanning electron microscopy (SEM) analysis. To evaluate the effects of luminal surface structure of decellularized aorta on thrombus formation and cell behavior, blood clotting tests and recellularization of endothelial cells and smooth muscle cells were performed. The results of the blood clotting test showed that the closer the luminal surface structure is to the native aorta, the higher the anti-coagulant property. The results of the cell seeding test suggest that vascular cells recognize the luminal surface structure and regulate adhesion, proliferation, and functional expression accordingly. These results provide important factors for preparing decellularized cardiovascular prostheses and will lead to future developments in decellularized cardiovascular applications.

## Introduction

Cardiovascular disease is one of the leading causes of death worldwide [1,2]. Due to a shortage of available donor organs and limitations of current artificial cardiovascular prostheses, artificial heart valves and vessels with anti-thrombotic and anti-infectious properties, high durability, growth potential, and no need for re-replacement surgery are desired.

**Funding:** This work was supported in part by a Grant-Aid for Scientific Research (B) (16H03180) from JSPS, the Creative Scientific Research of the Viable Material via Integration of Biology and Engineering from MEXT and the Cooperative Research Project of Research Center for Biomedical Engineering from MEXT.

**Competing interests:** The authors have declared that no competing interests exist.

Recently, decellularized tissues, which are the extracellular matrix obtained by removing cellular components from living tissues, have been widely developed. They have attracted increasing interest for a variety of applications in regenerative medicine [3]. Although the development of decellularized cardiovascular tissues has been studied [4–6], few have been clinically applied as of yet [3]. One of the reasons for this is that the mechanisms of decellularized tissues enabling a high biocompatibility and functionality are not yet fully understood.

Previously, we reported that decellularized aorta prepared by high-hydrostatic pressurization (HHP) showed good *in vivo* performance, including early reendothelialization and anti-thrombogenicity [7]. The HHP method disrupts the cells inside the tissue, and the cell debris can be removed by a series of washing processes without use of any surfactants. To determine the factors related to good *in vivo* performance, we previously evaluated mechanical properties and protein permeability, and it was found that HHP decellularized aortas have properties and functions similar to those of native aortas [8,9]. Furthermore, observation of the luminal surface structure showed that the luminal surface structure of HHP decellularized aorta was maintained, in contrast to other decellularized aortas prepared by different methods. Therefore, we hypothesized that vascular endothelial cell recruitment and anti-thrombogenicity occur in HHP-treated aortas because their luminal surface structure is maintained. Thus, the purpose of this study was to clarify the effects of the luminal surface structure on thrombus formation and vascular cell behavior using HHP decellularized aortas. To prepare decellularized aortas with various luminal surface structures, the luminal surface of decellularized aortas was disrupted by heating, drying, and peeling. The luminal surface structure was evaluated by immunohistological staining, collagen hybridizing peptide (CHP) staining, and scanning electron microscopy (SEM) analysis. To evaluate the effects of luminal surface structure of decellularized aorta on thrombus formation and cell behavior, blood clotting tests and recellularization of endothelial cells and smooth muscle cells were performed.

## Materials and methods

### Materials

Fresh porcine aortas and blood were obtained from a local slaughterhouse (Tokyo Shibaura Zouki, Tokyo, Japan). DNase I (10104159001) was purchased from Roche Diagnostics (Tokyo, Japan). Magnesium chloride hexahydrate ($MgCl_2 \cdot 6H_2O$) (135–15055), sodium chloride (NaCl), phosphate-buffered saline (PBS) (164–23551), neutral buffered (pH 7.4) solution of 10% formalin (062–01661), tert-butyl alcohol (028–03386), and proteinase-K (169–21041) were purchased from FUJIFILM Wako Pure Chemical Corp. (Osaka, Japan). Ethanol (99.5%, 14033–80) was purchased from Kanto Chemical Co., Inc. (Tokyo, Japan). Glutaraldehyde (25%, G004) was obtained from TAAB Laboratories Equipment, Ltd. (Berkshire, England) and Paraplast Plus® (P3683-1KG) was purchased from Sigma (St. Louis, MO, USA). Phenol/chloroform (311–90151) and ISOGEN with a spin column (318–07511) were purchased from Nippon Gene (Tokyo, Japan). Anti-collagen IV (1340–01) was obtained from SouthernBiotech (Alabama, USA). The collagen-hybridizing peptide, 5-FAM conjugate (F-CHP) (FLU60) was purchased from 3 Helix, Inc. (Utah, USA). Quant-iT™ PicoGreen™ dsDNA Assay Kit (P7581), Lambda DNA (#SD0011), and ProLong Gold Antifade Mountant with 4',6-diamidino-2-phenylindole (DAPI) (P36935) were obtained from Thermo Fisher Scientific K.K. (Tokyo, Japan). Human umbilical vein endothelial cells (HUVECs) (C-12208), human aortic smooth muscle cells (HAoSMCs) (C-12533), and smooth muscle cell growth medium 2 kit (C-22162) were purchased from Takara Bio (Shiga, Japan). EGM™-2 BulletKit™ (CC-3162) was obtained from Lonza (Chiba, Japan). Calcein-AM solution (148504-34-1) was purchased from Dojindo

Laboratories (Kumamoto, Japan). THUNDERBIRD$^{®}$ SYBR qPCR Mix (QPS-201) was purchased from Toyobo (Osaka, Japan).

## Preparation of decellularized porcine aorta

Fresh porcine aortas were purchased from a local slaughterhouse and stored at 4˚C until use. The aortas were washed with saline, and the surrounding tissue and fat were trimmed. The trimmed aortas were then cut longitudinally. These aortas were circularly cut out with an inner diameter of 15 mm using a hollow punch. The disked aortas were packed in plastic bags with saline and hermetically sealed. After the cells were destroyed by hydrostatic pressurization at 1000 MPa and 30˚C for 10 min using a hydrostatic pressurization system (Dr. Chef, Kobelco, Tokyo, Japan), the samples were washed with DNase (0.2 mg/mL) and $MgCl_2$ (50 mM) in saline at 4˚C for 7 d, followed by a change of the washing solution to 80% ethanol in saline at 4˚C for 3 d, and then to saline at 4˚C for 3 d to remove cell debris in the tissues.

To prepare decellularized aorta with various luminal surface structures, some HHP decellularized aortas were placed in a sterile flask with saline and placed in a heating mantle at 90˚C for 1 h (hereinafter referred to as HHP 90˚C tunica-intima). Some of the aortas were placed on a sterile drape with the inner surface facing up and air-dried on a clean bench for 30 min (hereinafter called HHP 30 min dried tunica-intima). The inner membrane was peeled off of some decellularized aortas to expose the tunica media (HHP tunica media). To prepare tunica media, the outer membrane of the decellularized HHP aorta was held with tweezers, while the inner membrane was peeled off with another tweezer.

## DNA quantification

The decellularized aortas were freeze-dried and dissolved in lysis buffer containing 50 μg/mL proteinase- K, 50 mM Tris-HCl, 1% sodium dodecyl sulfate (SDS), 10 mM sodium chloride (NaCl), and 20 mM ethylenediaminetetraacetic acid (EDTA) at 55˚C overnight. DNA extraction and purification were performed using phenol/chloroform and ethanol precipitation. The residual DNA content in the native and decellularized tissues was quantified using Quant-iT PicoGreen dsDNA reagent against a λDNA standard curve (0–1000 ng/mL) using a microplate reader at an excitation of 480 nm and an emission of 525 nm (Cytation 5, BioTek Instruments, Inc., Vermont, USA). The measurements were normalized to a tissue dry weight of 20 mg.

## Histological evaluation of decellularized aortas

The native aorta and decellularized aortas were fixed by immersion in a neutral-buffered (pH 7.4) solution of 10% formalin in PBS for 24 h at 25˚C and dehydrated in graded ethanol. The samples were then immersed in xylene and embedded in paraffin. The paraffin samples were cut into 4 μm-thick sections for H-E and 5 μm-thick sections for anti-type IV collagen staining.

## F-CHP staining

After paraffin was removed by rinsing with xylene and graded ethanol, all slides were rinsed twice with deionized water and three times with 1 × PBS to remove detergents. A solution containing 15 μm of F-CHP diluted with PBS was heated to 80˚C for 5 min to dissociate trimeric CHP into monomers. The heated CHP solution was cooled immediately in an ice bath to avoid thermal damage to the tissue sections. CHP solution was dropped onto each section, and the samples were incubated in a humidity chamber at 4˚C for 2 h. The slides were then washed three times with PBS before mounting.

## Scanning electron microscope (SEM) observation

A scanning electron microscope (S-4500/EMAX-700, Hitachi, Ltd., Tokyo, Japan) was used. The aortas were fixed with 2.5% glutaraldehyde in PBS and gradually dehydrated in ethanol. The dehydrated aortas were placed in tert-butyl alcohol and then vacuum-dried. Before observation, the surfaces of the decellularized aortas were coated with gold.

## Evaluation of thrombogenicity of decellularized aortas

Fresh porcine blood was obtained from a local slaughterhouse and stored at 4°C until use. Blood coagulation tests were performed according to a previously described protocol [10]. Briefly, whole blood containing 0.324% citric acid was prepared with a tenth of calcium chloride ($CaCl_2$) for the coagulation of blood for 15 min on glass (C022221, Matsunami Glass Ind., Ltd.). A stainless-steel tray covered with a moistened paper towel was floated in a 37°C water bath. Whole blood was dropped onto the cover glass lined up on the tray. Each cover glass was picked up at 2, 4, 6, 8, 10, and 15 min, washed with saline, and then checked for whether the whole blood had coagulated at 15 min. When the concentration of $CaCl_2$ was determined, the same amounts of $CaCl_2$ and whole blood were mixed. Glass, polytetrafluoroethylene (PTFE) (7-359-01, Flonchemical Co., Ltd., Oosaka, Japan), and disked decellularized aortas were placed on the tray in the same way. Fifty μl of whole blood containing $CaCl_2$ were dropped onto the samples, which were washed with saline at the same time intervals as described above, and photographs were taken. Glass, PTFE, and all decellularized aortas were then placed into a 24-well plate filled with deionized water, resulting in hemolysis. After 24 h, the absorbance at 576 nm of the hemolyzed blood was measured using an absorbance meter. The obtained absorbance measurements were substituted into the following equation for calculating the percentage of the hemoglobin content in each sample.

$$\text{Hemoglobin content } (\%)$$
$$= \frac{\text{Absorbance of each sample} - \text{Absorbance of deioinized water}}{\text{Absorbance of whole blood}} \times 100(\%)$$

## Cell seeding

The disked aortas (inner diameter 15 mm) were placed onto a 24-well tissue culture plate, and a stainless-steel ring with a 13 mm inner and 15 mm outer diameter was placed onto them to avoid curling. HUVECs were seeded with $2 \times 10^4$ cells/cm$^2$ and HAoSMCs were seeded with $1 \times 10^4$ cells/cm$^2$ on the surface of the decellularized aortas and incubated at 37°C under 5% $CO_2$ conditions for 7 d.

## Measurement of cell proliferation

HUVECs and HAoSMCs were stained with calcein-AM and then incubated for 30 min at 37°C. The cells were observed under a fluorescence microscope (BZ-X710, Keyence Corp., Osaka, Japan). The number of cells was counted in sections from all samples, and the cell density was calculated using the counted cell numbers.

## Quantitative reverse-transcription polymerase chain reaction (qPCR)

Total RNA was extracted from HAoSMCs on TCPS and decellularized aortas using ISOGEN with a spin column according to the manufacturer's instructions. GAPDH was used to normalize the gene expression. Quantitative PCR was performed using Power SYBR Green PCR Master Mix on a StepOnePlus system (Thermo Fisher Scientific K.K., Tokyo, Japan) with the Delta Delta Ct method. The forward and reverse primer sequences are shown in Table 1.

**Table 1. List of forward and reverse primer for reverse transcription polymerase chain reaction.**

| Gene sequence | Forward | Reverse |
|---|---|---|
| human GAPDH | GGAGCGAGATCCCTCCAAAAT | GGCTGTTGTCATACTTCTCATGG |
| human SMA | ATGAAGGATGGCTGGAACAG | GCGTGGCTATTCCTTCGTTA |
| human CNN | CATCTGCAGGCTGACATTGA | AGCTAAGAGAAGGGCGGAAC |
| human SM22 | CGGTAGTGCCCATCATTCTT | AACAGCCTGTACCCTGATGG |

## Statistical analysis

The quantitative analysis of residual DNA (Fig 1), F-CHP intensity (Fig 3(F)), and cell density (Figs 6 and 7) are expressed using the mean ± standard deviation (SD). The quantitative analysis of the blood coagulation rate data (Fig 5) and qRT-PCR (Fig 8) were expressed using the mean ± standard error of the mean (SE). The Student's $t$-test was used to determine significant differences (Fig 1). Tukey's multiple-comparison test was used to test for statistical significances in Figs 3(F), 5(B), 6(K), 7(K), 8(A) and 8(B).

## Results

Porcine aortas were decellularized, and the amount of residual DNA was measured (Fig 1). The amount of DNA remaining in the HHP-treated aorta was significantly lower than that in the untreated aorta.

Fig 2(A)–2(J) show H-E and Type IV collagen-stained native aortas and decellularized aortas. The efficacy of decellularization was verified using H&E staining. No nuclei were detected in sections with decellularized aortas after HHP treatment (Fig 2(B)–2(E)). The tunica intima of the HHP-treated aorta and the HHP 30 min dried tunica-intima were observed to be similar to that of the untreated aorta (Fig 2(B) and 2(D)). HHP 90˚C tunica intima had larger wave-like shapes and a flatter fibril structure than the native aorta (Fig 2(C)). In the tunica media of HHP-treated aorta, the gaps between collagen fibrils were enlarged due to the peeling of the tunica intima (Fig 2(E)). Immunostaining for type IV collagen was also performed. Type IV collagen is the main component of the basement membrane and forms networks that provide

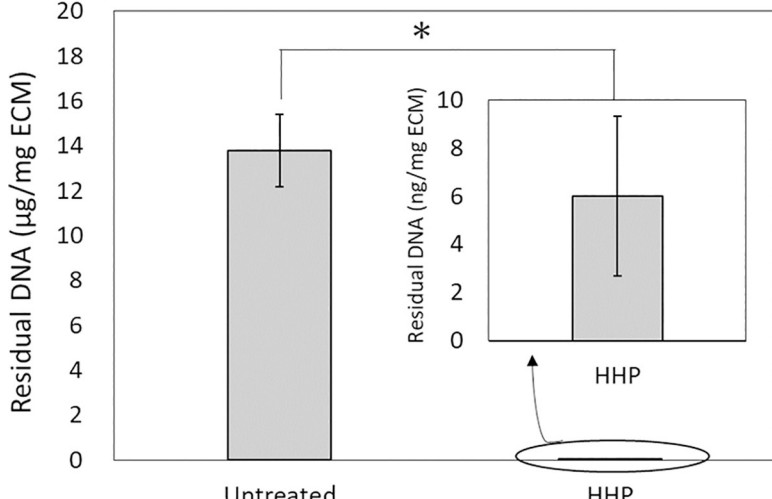

**Fig 1. Quantitative analysis of residual double stranded DNA (dsDNA) in untreated and decellularized aortas.**
$^*p < 0.001$.

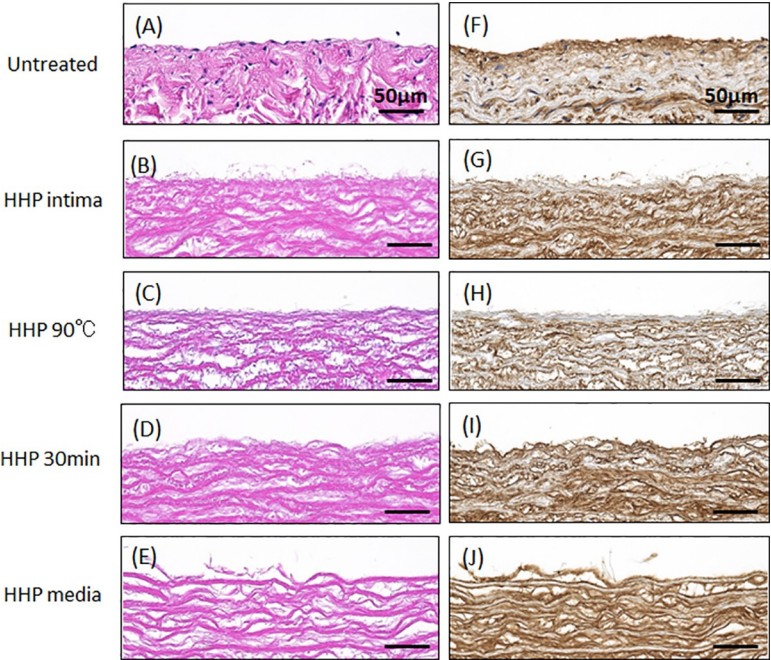

**Fig 2.** H-E staining (A-E), type IV collagen staining (F-J) of untreated and decellularized aortas. (A)(F) Untreated aorta, (B)(G) HHP tunica-intima, (C)(H) HHP 90°C tunica-intima, (D)(I) HHP 30 min dried tunica-intima, (E)(J) HHP tunica-media. Scale bar: 50 μm.

structural support for endothelial cells. The type IV collagen layer was preserved in HHP tunica intima, even after drying (Fig 2(G) and 2(I)). HHP 90°C tunica intima was weakly stained, so it is assumed that type IV collagen was denatured by the heating treatment (Fig 2 (H)).

CHP staining, which specifically targets denatured collagen chains, was performed (Fig 3 (A)–3(E)), and the fluorescence signals in the scanned images were quantified (Fig 3(F)). As

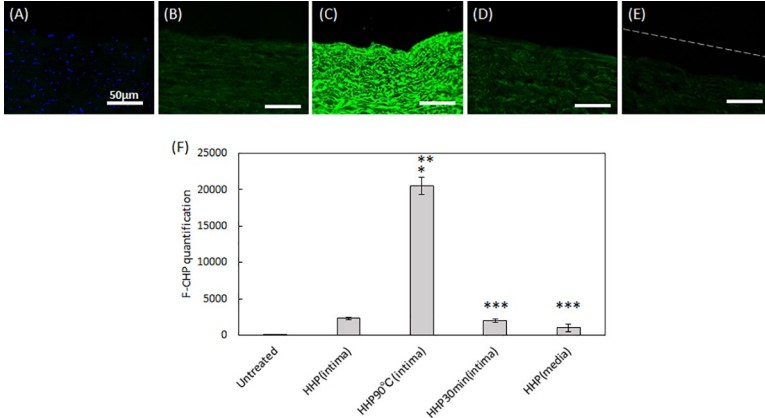

**Fig 3. Fluorescence images showing F-CHP staining on sections.** (A) untreated aorta, (B) HHP tunica-intima, (C) HHP 90°C tunica-intima, (D) HHP 30 min dried tunica-intima, (E) HHP tunica-media. Scale bar: 50 μm. (F) integrated F-CHP signals quantified from images of the tissue sections. The asterisk (*) indicates statistical significance in comparison with the untreated aorta ($p < 0.01$). Two asterisks (**) indicate statistical significance in comparison with HHP tunica-intima ($p < 0.01$). Three asterisks (***) indicate statistical significance in comparison with HHP 90°C tunica-intima ($p < 0.01$).

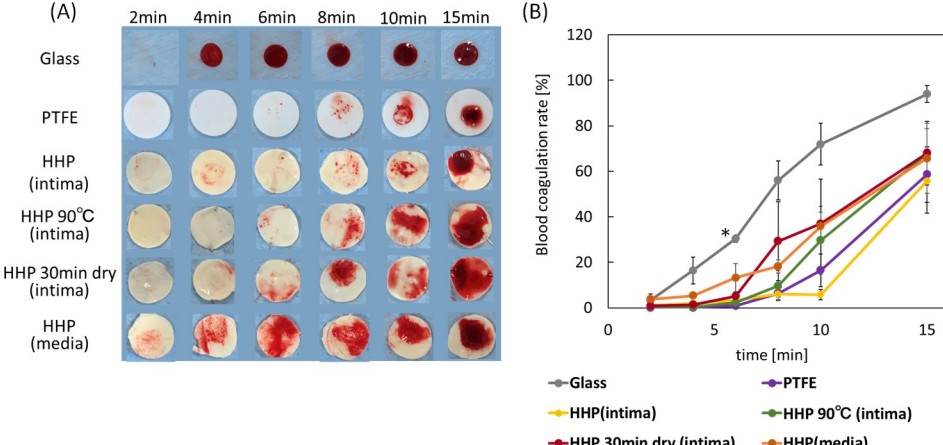

**Fig 4. SEM observation of surface.** (A)(F) Untreated aorta, (B)(G) HHP tunica-intima, (C)(H) HHP 90°C tunica-intima, (D)(I) HHP 30 min dried tunica-intima, (E)(J) HHP tunica-media. Scale bar: (A)–(E) 10 μm, (F)–(J) 500 μm.

expected, no fluorescence intensity except for DAPI was detected in the untreated aorta (Fig 3 (A) and 3(F)). HHP tunica intima exhibited a low intensity signal, suggesting slightly denatured collagen (Fig 3(B) and 3(F)). In HHP 90°C tunica intima, a strong fluorescence signal was detected, indicating that the collagen was completely destroyed by the heating process (Fig 3(C) and 3(F)). No significant differences were observed between the fluorescence image and the intensity signal of the HHP tunica intima and that of the HHP 30 min dried tunica intima and HHP tunica media (Fig 3(D)–3(F)).

SEM was used to analyze the fiber structures of the decellularized aortas (Fig 4). The fibers were observed in magnified 3.0k SEM images (Fig 4(A)–4(E)), while the surfaces were observed in 100 SEM images (Fig 4(F)–4(J)). For HHP-treated tunica intima, the fiber bundles were oriented longitudinally and similar to in the untreated aortas (Fig 4(B) and 4(G)). As for the HHP 90°C tunica-intima, shrunken and wavy fibers were observed in the magnified images compared with HHP tunica-intima (Fig 4(C)), and smooth plane surfaces without ruggedness are shown in Fig 4(H). As for the HHP 30 min dried tunica-intima, there was not much difference in fiber structure compared to HHP tunica intima; however, numerous fine cracks were observed on the surface of HHP 30 min dried tunica-intima (Fig 4(I)). The HHP-treated tunica media exhibited a rough surface due to the peeling of the tunica intima (Fig 4(E) and 4(J)). Based on the above observations, various luminal surface structures of decellularized aortas were prepared.

**Fig 5.** (A) Photograph of blood coagulation times of decellularized aortas with various basement membrane structures. (B) Blood coagulation rate of decellularized aortas. Error bars represent S.E. Asterisk (*) shows a significant difference (*p* < 0.05) between glass and PTFE, HHP (intima), HHP 90°C (intima), HHP 30 min dry (intima) at 6 min.

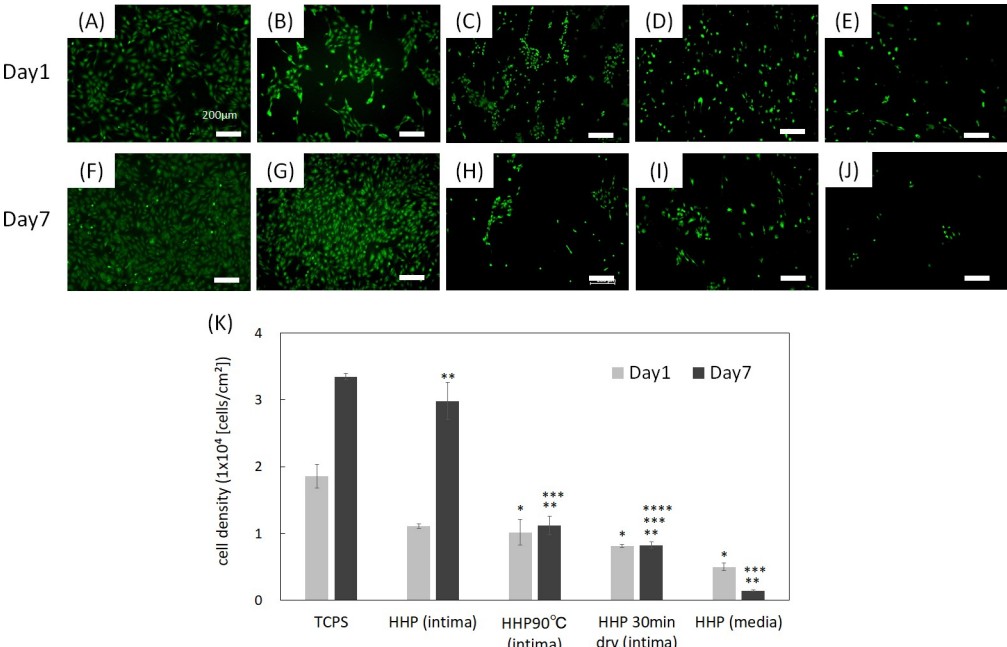

**Fig 6.** Fluorescence images of HUVECs at day 1 and day 7 on (A), (F) TCPS, (B), (G) HHP tunica-intima, (C), (H) HHP 90˚C tunica-intima, (D), (I) HHP 30 min dried tunica-intima, (E), (J) HHP tunica-media. Scale bar: 200 μm. (K) Cell density. The cell density of each sample was compared each day. Asterisk (*) indicates statistical significance in comparison with TCPS on day 1 ($p < 0.05$). Two asterisks (**) indicate statistical significance in comparison with TCPS on day 7 ($p < 0.05$). Three asterisks (***) indicate statistical significance in comparison with HHP tunica-intima on day 7 ($p < 0.05$). Four asterisks (****) indicate statistical significance in comparison with HHP 90˚C tunica-intima on day 7 ($p < 0.05$).

In this study, the Lee-White test was used to examine the effects of luminal surface structure on the thrombogenicity of decellularized aortas. Fig 5(A) shows an *in vitro* evaluation of a blood clotting test for decellularized aortas. Glass and PTFE were used as the negative and positive controls, respectively. On glass, a blood clot formed at 4 min, while no clot formation occurred until 15 min on PTFE. As for the HHP-treated tunica-intima, no clot formation was observed until 15 min, which is almost the same result as for PTFE. As for HHP 90˚C tunica intima, HHP 30 min dried tunica intima, and HHP treated tunica media, thrombus formation occurred within 10 min. The result in Fig 5(B) shows the measurement of the absorbance of the hemolyzed blood clot. The difference between the decellularized samples and PTFE was not statistically significant; in short, all HHP decellularized aortas showed a high anti-coagulability. As shown in Fig 5(B), however, HHP tunica intima showed the highest anti-coagulability among all samples at each time point.

It is widely known that endothelial cell coverage on the luminal surface of cardiovascular grafts immediately after transplantation is required for preventing thrombus formation [11,12]. Endothelial cells also regulate vessel tone, platelet and leukocyte activation, and SMC migration and proliferation [13]. In this study, HUVECs were chosen for evaluating the recellularization efficacy and cell behavior of decellularized aortas with various luminal surface structures. The attachment and proliferation of HUVECs on decellularized aortas were observed using a fluorescence microscope. The number of cells was then counted in sections from all samples, and the cell density was calculated using the counted cell numbers (Fig 6 (K)). As can be seen in the results, HUVECs adhered on all samples. HUVECs proliferated well on HHP-treated tunica intima (Fig 6(B), 6(G) and 6(K)). The attachment of HUVECs on

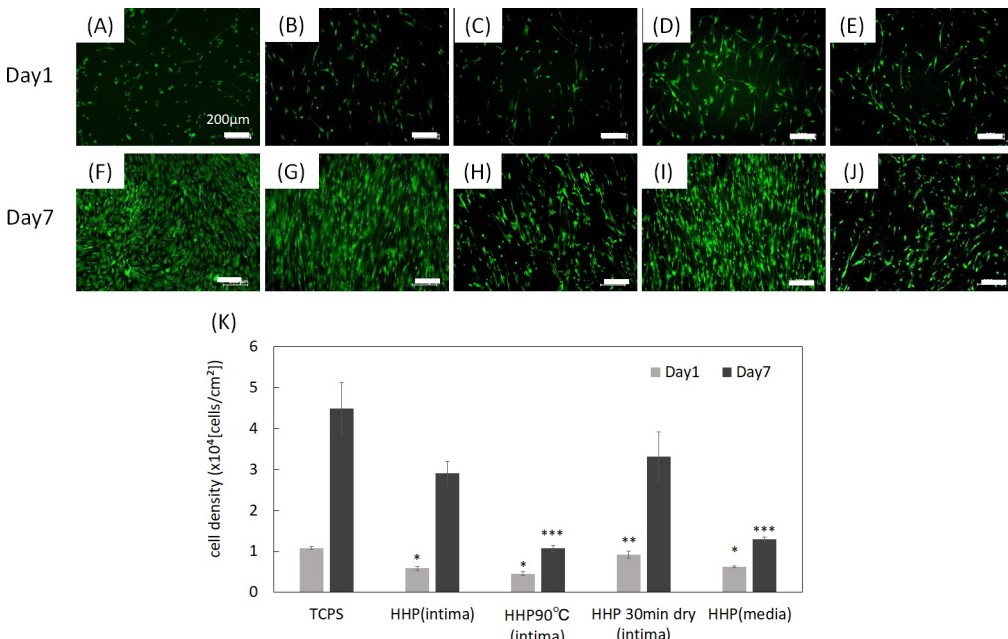

**Fig 7.** Fluorescence images of HAoSMCs on day 1 and day 7 on (A), (F) TCPS, (B), (G) HHP tunica-intima, (C), (H) HHP 90˚C tunica-intima, (D), (I) HHP 30 min dried tunica-intima, (E), (J) HHP tunica-media. Scale bar: 200 μm. (K) Cell density. The cell density of each sample was compared each day. Asterisk (*) indicates statistical significance in comparison with TCPS on day 1 ($p < 0.05$). Two asterisks (**) indicate statistical significance in comparison with HHP 90˚C tunica-intima on day 1 ($p < 0.05$). Three asterisks (***) indicate statistical significance in comparison with TCPS on day 7 ($p < 0.05$).

HHP 90˚C tunica-intima and HHP 30 min dried tunica-intima were nearly equal to that of the HHP treated tunica intima, but the morphology of the cells was different. HUVECs on HHP tunica-intima extended, while HUVECs on HHP 90˚C and HHP 30 min dried tunica-intima did not extend and showed a round shape (Fig 6(C), 6(D), 6(H) and 6(I)). It is considered that HUVECs were shed during cell culture, resulting in little cell attachment on HHP 90˚C and HHP 30 min dried tunica-intima on day 7 (Fig 6(K)). As for the HHP treated tunica media, the initial adhesion of HUVECs was low, and the number of HUVECs decreased during the 7 days of cultivation (Fig 6(E), 6(J) and 6(K)).

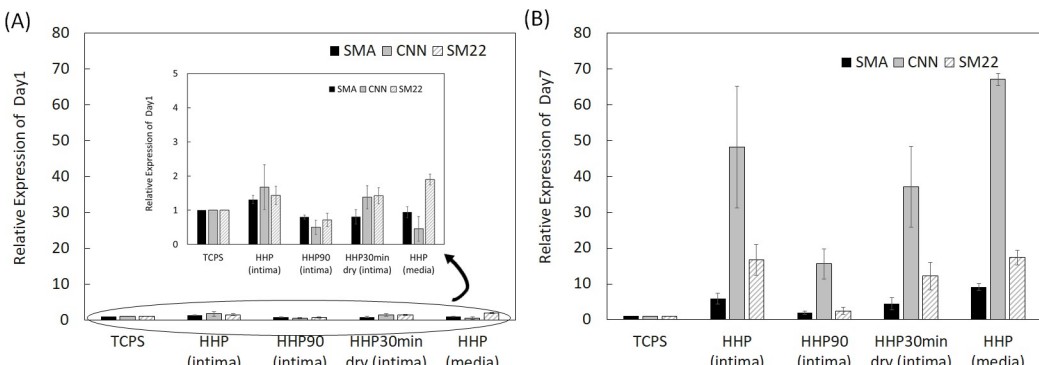

**Fig 8.** mRNA expression of SMA, CNN and SM22 in HAoSMCs on TCPS and decellularized aortas, (A) on day 1 and (B) on day 7. The relative expression of each gene was compared with all samples for statistical analysis.

Vascular smooth muscle cells (VSMCs) are present in the tunica media (middle layer) and have secretory ability of the major extracellular proteins, including collagen, elastin, and proteoglycans, all of which influence the mechanical properties [14]. Furthermore, VSMCs can switch between contractile and synthetic phenotypes in response to changes in the local environment [15,16]. They usually exhibit a contractile phenotype and express contractile proteins such as $\alpha$-smooth muscle actin (SMA), smooth muscle calponin (CNN) and SM22 $\alpha$ (SM22). In the contractile phenotype, they have a low proliferative rate to maintain the ECM of the tunica media [15]. In response to vascular injury, including tissue damage, they alter their phenotype to a synthetic state and reduce the expression of contractile proteins, with increased proliferation and remodeling of the ECM to facilitate migration [17–19]. In this study, the effects of the luminal surface structure of the decellularized aorta on VSMC proliferation and phenotype were investigated. Human aorta smooth muscle cells (HAoSMCs) were seeded on TCPS and decellularized aortas. The cell density was calculated by counting the number of cells on each sample (Fig 7(K)). As shown in Fig 7(A)–7(J), HAoSMCs adhered on all samples. HAoSMCs proliferated on HHP-treated tunica-intima and HHP 30 min dried tunica intima (Fig 7(B), 7(D), 7(G), 7(I) and 7(K)). The initial attachment of HAoSMCs onto HHP 90°C tunica-intima and HHP tunica media was nearly equal to that of HHP tunica intima, but the cells did not proliferate (Fig 7(C), 7(E), 7(H), 7(J) and 7(K)).

The expression levels of SMA, CNN, and SM22 from HAoSMCs on the samples, which were expressed in the contractile phenotype, were examined by qRT-PCR analysis. On day 1, HAoSMCs on all samples showed a low gene expression (Fig 8(A)). In contrast, on day 7, the expression ratio of the contractile phenotype genes was high, except for tunica intima heated at 90°C (Fig 8(B)).

## Discussion

To identify the effect of the luminal surface structure of the decellularized aorta on thrombus formation and cell behavior, porcine aortas were decellularized using the HHP method, which has been previously reported to have good *in vivo* performance [7]. Their luminal surfaces were then processed by heating, drying, and peeling. The luminal surface structure and components were evaluated using type IV collagen immunostaining, CHP staining, and SEM analysis. Type IV collagen immunostaining was performed to evaluate the maintenance of the basement membrane of the decellularized aorta. HHP decellularization did not affect type IV collagen, but type IV collagen was not stained after the heating process. This result indicates that type IV collagen was denatured and gelatinized during the heating process to 90°C. CHP staining is a method used to examine the unfolding of collagen molecules in decellularized tissues [20]. Compared with the untreated aorta, slightly unfolded collagen was observed when aortas were decellularized using the HHP method. The image and quantification analysis of HHP 90°C heated tunica intima clearly showed the denaturation of collagen molecules. SEM observation was performed to observe the fiber structure of the decellularized aorta. In the HHP 90°C heated tunica intima, the fibers appeared shrunken and wavy because the collagen fibers had been denatured by heat treatment. This result correlates with a previous report which showed that collagen fibers shrink, and that their fiber diameters increase when they are treated with temperatures at or above 50°C [21,22].

The effect of the luminal surface of the decellularized aorta on thrombus formation was assessed by a blood clotting test. The results showed that the anti-coagulability was higher for HHP-treated tunica-intima and lower for decellularized aortas with disordered luminal surface structures (HHP 90°C tunica-intima, HHP 30 min dried tunica-intima, HHP tunica-media). Previously, it was reported that damaging the luminal surface of grafts by drying

decreased graft patency after transplantation [23]. Therefore, this *in vitro* result correlates with the *in vivo* results and indicates that the more similar the luminal surface structure is to the native aorta, the higher the anti-coagulant property.

The effect of the luminal surface of the decellularized aorta on cell behavior and gene expression was evaluated. Endothelial cells and smooth muscle cells are the main cellular components of the aorta and interact with each other to maintain the function and mechanical properties of the vessel [24]. It is important to evaluate whether these vascular cells, which play essential roles *in vivo*, recognize the place where they should exist and proliferate and express their functions properly. HUVECs and HAoSMCs were seeded on decellularized aortas, and the phenotype of HAoSMCs was evaluated by qRT-PCR. Regarding the qRT-PCR results on day 1, it was assumed that the gene expression ratio was low in all samples because the phenotype of the adhered HAoSMCs could not have switched their phenotype in such a short period. HUVECs adhered and proliferated on HHP decellularized tunica-intima because the luminal surface structure and basement membrane were well maintained. HAoSMCs showed high proliferation as they recognized the luminal surface as tunica intima. This cell behavior seems to be correlated with the event of neointima hyperplasia occurring, because the population of SMCs migrates from the media and proliferates within the intima during neointima formation [25,26]. However, the expression of HAoSMC contractile phenotype marker genes was detected in the results of qRT-PCR on day 7. It is assumed that the cell density exceeded the capacity of the luminal surface, so that HAoSMCs shifted the synthetic phenotype to the contractile phenotype. As for the HHP 90˚C heated tunica intima, in which luminal surface structure and collagen were denatured, initial adhesion of both HUVECs and HAoSMCs was low and did not proliferate. This may be because the luminal surface structure was disordered, and type IV collagen was denatured by the heating process. As for the HHP 30 min dried tunica intima, in which the luminal surface structure was cracked by drying, the number of attached HUVECs during cell culturing was low, while HAoSMCs adhered well and proliferated. Based on this result, it was suggested that drying out the luminal surface of the decellularized aorta may lead to neointima hyperplasia. This *in vitro* result correlates with a previous *in vivo* study that showed vessel occlusion when the intima were dried before implantation [23]. As for the HHP tunica media, initial adhesion of HUVECs was observed, but they were shed during cell culturing. This may be because the basement membrane, which supports endothelial cells, was removed. HAoSMCs cultured on HHP tunica media did not proliferate. This phenomenon is similar to the behavior of vascular smooth cells *in vivo* for which it has been reported that smooth muscle cells within media exhibit the stable contractile type with low proliferative capacity [15]. Thus, HAoSMCs recognized the luminal surface structure and component of tunica media and exhibited a stable contractile type with low proliferative capacity *in vitro*. These results suggest that HUVECs and HAoSMCs recognize the luminal surface and component of the decellularized aorta and proliferate and/or express their functions properly. Therefore, good maintenance of the luminal surface structure of the decellularized aorta so that it is similar to that of the native aorta is one of the key factors in preparing decellularized cardiovascular applications.

## Conclusion

In the present study, decellularized aortas with different luminal surface structures were prepared to investigate the effects of their structures on thrombus formation and cell behavior. The results of the blood clotting test showed that the more similar the luminal surface structure is to that of the native aorta, the higher the anti-coagulant property. From the results of cell seeding and qRT-PCR, it was found that endothelial cells and smooth muscle cells recognized

the luminal surface structure and components of the decellularized aorta and regulated adhesion, proliferation, and functional expression. These results provide useful insights into future developments of decellularized cardiovascular applications.

## Supporting information

**S1 File. Quantitative analysis of residual dsDNA.**
(XLSX)

**S2 File. F-CHP intensity.**
(XLSX)

**S3 File. Blood coagulation rate.**
(XLSX)

**S4 File. HUVEC cell density.**
(XLSX)

**S5 File. HAoSMC cell density.**
(XLSX)

**S6 File. qRT-PCR_HAoSMC day 1.**
(XLSX)

**S7 File. qRT-PCR_HAoSMC day7.**
(XLSX)

## Author Contributions

**Conceptualization:** Mako Kobayashi, Akio Kishida.

**Funding acquisition:** Akio Kishida.

**Investigation:** Mako Kobayashi, Masako Ohara.

**Methodology:** Yoshihide Hashimoto, Tsuyoshi Kimura.

**Project administration:** Naoko Nakamura, Akio Kishida.

**Supervision:** Naoko Nakamura, Toshiya Fujisato, Akio Kishida.

**Writing – original draft:** Mako Kobayashi, Masako Ohara.

**Writing – review & editing:** Yoshihide Hashimoto, Tsuyoshi Kimura, Akio Kishida.

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
