## [Decision Letter · Decision Letter 0]

10 Feb 2021

PONE-D-21-01350

Effect of luminal surface structure of decellularized aorta on thrombus formation and cell behavior

PLOS ONE

Dear Dr. Kishida,

Thank you for submitting your manuscript to PLOS ONE. After careful consideration, we feel that it has merit but does not fully meet PLOS ONE’s publication criteria as it currently stands. Therefore, we invite you to submit a revised version of the manuscript that addresses the points raised during the review process.

Please carefully check the comments from reviewers and make a revision to satisfy it. I will add an additional reviewer to correctly judge your revision.

We look forward to receiving your revised manuscript.

Kind regards,

Michiya Matsusaki

Academic Editor

PLOS ONE

Journal Requirements:

Reviewers' comments:

Reviewer's Responses to Questions

**Comments to the Author**

1. Is the manuscript technically sound, and do the data support the conclusions?

Reviewer #1: Yes

Reviewer #2: No

2. Has the statistical analysis been performed appropriately and rigorously? 

Reviewer #1: Yes

Reviewer #2: N/A

3. Have the authors made all data underlying the findings in their manuscript fully available?

Reviewer #1: Yes

Reviewer #2: Yes

4. Is the manuscript presented in an intelligible fashion and written in standard English?

Reviewer #1: Yes

Reviewer #2: No

5. Review Comments to the Author

Reviewer #1: Dear Authors,

(Ms. No.; PONE-D-21-01350)

In this study, the authors investigate the thrombus formation and the in vitro behaviors of vascular cells on decellularized aorta focusing on the surface structure. The in vitro results of your study are indicating that vascular events should be affected by the surface structure of decellularized aorta. This finding will give us useful knowledge for designing suitable decellularized tissue for clinical applications

I would like to suggest the authors to clarify the following criticisms.

1) Introduction, line 53 - 56

Could you please add the details of why you hit upon your hypothesis focusing on the luminal surface structure of the HHP treated aorta. Many conceivable factors might affect the biological responses to HHP treated aortas.

I recommend making the distinction between molecular structure and tissue structure.

2) 2.1. Preparation of decellularized porcine aorta

Please describe more details about the treatments after decellularization. Do you have any reason why you decided 90 degrees for heating treatment? Did you dry decellularized aorta in the air for 30 min? How did you peel off the inner membrane from the decellularized aorta with?

3) 2.3. Histological evaluation of decellularized aortas

My concern is that the reagents used in the process to make the section for collagen staining might affect the structure of the collagen molecule. Is the collagen molecule not denatured by xylene, ethanol, and paraffin? Is the frozen section not suitable for this experiment?

4) 2.6. Evaluation of thrombogenicity of decellularized aortas

What from did you obtain the whole blood for this evaluation? Please add the information about approvals if you obtained the whole blood from human donors or animals.

5) 2.7 Cell seeding

Please indicate the size of decellularized aortas. I think that TAKARA Bio Inc. is located in Shiga prefecture, not Tokyo.

6) 2.9. qPCR

The authors forgot to indicate the Table of primers. There is no Table in the manuscript. I previously make an apology if I overlooked it.

7) 2.10. Statistical analysis

As you know, the one-way ANOVA is suitable for evaluating the significant differences among more than three groups.

8) 3. Results and Discussion, line 154

Please remove “Discussion” because you subsequently prepared “4. Discussion”.

9) 3. Results, line 210-211

“Glass and PTFE were used as negative and positive controls, respectively.”

Is glass a positive control, right? I think the position between Glass and PTFE should be placed opposite. Please add the further details (manufacture and name) of the Glass and PTFE you used.

10) 3. Results, line 226 – 227

“It is widely known that endothelial cell seeding on luminal surface of cardiovascular grafts are required to prevent thrombus formation (9, 10).”

This sentence is complicated for me. What did you mean by “cell seeding”? Is “cell seeding” meaning that cells are seeded on cardiovascular grafts ex vivo?

Also, “,,,,,grafts are” should be corrected to “,,,,,,grafts is” in this sentence.

11) Figure 3

Were the parameters of CLSM fixed for the observation of all samples? How did you adjust and calibrate the fluorescence intensity for all observations? The fluorescence intensity should be calibrated because the tissue has self-fluorescence and you can control the fluorescence intensity by changing the parameter for CLSM observation.

I recommend doing the quantitative evaluation of collagen denaturation using the fluorescence intensity of these pictures.

12) Fig. 6

I agree that both types of cells recognized “something” on decellularized aortas and the “something” is relating to the denaturation of the collagen molecule. But, I cannot believe that cells directly recognize the denatured collagen because the amino acid sequence of collagen did not change by the denaturation process. The initial adhesion of HUVECs is similar between HHPi and HHPi-90 degrees, but the proliferation behavior of adherent HUVECs was completely different. This means that HUVECs recognize the adhesive sites in collagen molecule on both aortas as well. Then, HUVECs was affected by “something” and was not able to proliferate on HHPi-90 degrees. Do you have any speculation about “something”? Was the tissue structure of HHPi-90 degrees maintained during cell culture for 7 days?

13) Fig. 8

Please add the enlarged view in Fig. 8 (A). It was so difficult to recognize a difference in the gene expression level of all samples. Please perform the statistical analysis. In my opinion, you can remove this figure if you do not use it for discussion.

14) 4. Discussion, line 314-315

“Cell seeding on decellularized aortas and the evaluation of mRNA expression of those cells by RT-PCR were performed.”

Does “those cells” mean both HUVECs and AOSMCs? If so, the gene expression analysis of HUVECs should be added to this manuscript.

15)

There are several incomprehensible sentences and grammatical errors in this manuscript. [For example, line 304-308 and 310-314]

I recommend that this manuscript should be checked carefully or get the English proofreading.

Reviewer #2: The manuscript studies the effect of luminal surface structure on thrombus formation and cell behavior using HHP decellularized aortas. The luminal surface structure was disrupted by heating, drying and peeling. The resultant surfaces were analyzed by immunohistological staining, collagen hybridizing peptide (CHP) staining and scanning electron microscopy (SEM) analysis. Overall, the manuscript has a poor-written language and requires major revision in several parts. It lacks novelty, and the data is not interesting. Here are some points that may improve the quality of the manuscript:

1) The lines from 59 to 66 are copied from the abstract.

2) Information (company-cat number for antibody) for chemicals reagents and instruments need to be mentioned in the materials and methods.

3) How does the inner membrane peel off? explain more on methods.

4) Figures legends should not be in the manuscript.

5) Why did the Author investigate the binding and proliferation of SMC in the lumen? These cells reside in medial layer.

6) Scale bar in figure 2 is absent.

7) CF-CHP staining images in Figure 3 have different contrast, limiting meaningful comparison between images.

8) In line 197, the authors mentioned that SEM analysis showed thicker fibers of HHP 90 ºC tunica-intima. Quantification of fiber diameters from SEM images among the samples should be provided with statistical analysis.

9) In line 199, from Fig. 4D and I, the authors indicated more gap between fibers on HHP 30 min dried tunica intima. The gap distance among the samples has to be quantified and analyzed statistically.

10) In line 226, the paragraph has only one sentence.

11) The images in figure 4G-I have different contrast compared with other images in the same figure.

12) Statistical analysis in Figures 6K, 7K, and 8 is absent.

13) The manuscript has grammatical errors, several too-short sentences, and the language needs revision.

---

## [Author Response · Author response to Decision Letter 0]

30 Mar 2021

Response to reviewers

Title: Effect of luminal surface structure of decellularized aorta on thrombus formation and cell behavior.

Thank you for your information together with feedback from the reviewers of the above paper. We have read reviewer’s comments very carefully. We have revised the manuscript according to the comments and suggestions of reviewers.

Reviewer #1: Dear Authors,

(Ms. No.; PONE-D-21-01350)

In this study, the authors investigate the thrombus formation and the in vitro behaviors of vascular cells on decellularized aorta focusing on the surface structure. The in vitro results of your study are indicating that vascular events should be affected by the surface structure of decellularized aorta. This finding will give us useful knowledge for designing suitable decellularized tissue for clinical applications

I would like to suggest the authors to clarify the following criticisms.

Answer:

We appreciate your kind comment. We response the reviewer’s questions point by point as below.

1) Introduction, line 53 - 56

Could you please add the details of why you hit upon your hypothesis focusing on the luminal surface structure of the HHP treated aorta. Many conceivable factors might affect the biological responses to HHP treated aortas.

I recommend making the distinction between molecular structure and tissue structure.

Answer: 

We have previously reported that reendothelialization and high patency were observed in HHP decellularized aorta, when decellularized aortas prepared by different decellularization methods were implanted in vivo (1). To find the factors related to good in vivo performance, we previously evaluated mechanical properties and protein permeability, and it was found that HHP decellularized aortas have similar properties and functions similar to those of native aortas (2), (3). Furthermore, observation of the luminal surface structure showed that structure of HHP decellularized aorta was maintained, in contrast to other decellularized aortas prepared by different methods. Therefore, we proposed hypothesis that vascular endothelial cell recruitment and anti-thrombogenicity occur on the HHP treated aorta since its luminal surface structure on aorta is well maintained. Thus, we prepared HHP decellularized aortas with different surface structure and evaluated the effects on vascular cell behavior.

 We added the reason why we hit upon the hypothesis in P.4, line 57-62.

Reference 

1) Funamoto S, Nam K, Kimura T, Murakoshi A, Hashimoto Y, Niwaya K, et al. The use of high-hydrostatic pressure treatment to decellularize blood vessels. Biomaterials. 2010;31(13): 3590-3595. DOI: 10.1016/j.biomaterials.2010.01.073.

2) Wu PL, Kimura T, Tadokoro H, Nam K, Fujisato T, Kishida A. Relation between the tissue structure and protein permeability of decellularized porcine aorta. Mater Sci Eng C Mater Biol Appl. 2014;43: 465-471. DOI: 10.1016/j.msec.2014.06.041.

3) Wu PL, Nakamura N, Kimura T, Nam K, Fujisato T, Funamoto S, et al. Decellularized porcine aortic intima-media as a potential cardiovascular biomaterial. Interact Cardiovasc Thorac Surg. 2015;21(2): 189-194. DOI: 10.1093/icvts/ivv113.

2) 2.1. Preparation of decellularized porcine aorta

Please describe more details about the treatments after decellularization. Do you have any reason why you decided 90 degrees for heating treatment? Did you dry decellularized aorta in the air for 30 min? How did you peel off the inner membrane from the decellularized aorta with?

Answer:

Since it is known that collagen denatures at temperatures above 50°C, we firstly tried to heat samples 

at 60°C which was insufficient for collagen denaturation at fiber structure and CHP staining. So, we 

set the temperature at 90°C to completely damage collagen fibers. As for the 30 min dried tunica-intima, disked HHP decellularized tunica-intima were air-dried in a clean bench for 30 min with the lumen surfaces upward. To prepare tunica media, the outer membrane of the decellularized HHP aorta was held with tweezer while inner membrane was peeled off with another tweezer. 

We added the detailed methods of preparing HHP 30 min dried decellularized aorta and HHP tunica-media in P.7, line 111-112 and 114-116.

3) 2.3. Histological evaluation of decellularized aortas

My concern is that the reagents used in the process to make the section for collagen staining might affect the structure of the collagen molecule. Is the collagen molecule not denatured by xylene, ethanol, and paraffin? Is the frozen section not suitable for this experiment?

Answer:

 Since no fluorescence intensity except DAPI was detected in untreated aorta from the results of F-CHP staining, we believe that collagen molecule was not denatured by xylene, ethanol and paraffin. We consider that this is because both untreated aorta and decellularized aortas were fixed using glutaraldehyde before staining. 

4) 2.6. Evaluation of thrombogenicity of decellularized aortas

What from did you obtain the whole blood for this evaluation? Please add the information about approvals if you obtained the whole blood from human donors or animals.

Answer:

　We apologize for not specifying it in the manuscript. Porcine whole blood was purchased from local slaughterhouse. This product is for research experiment and no need approvals. We added “Materials” in the “Materials and Methods” section in P.5-6, line 75-96. 

5) 2.7 Cell seeding

Please indicate the size of decellularized aortas. I think that TAKARA Bio Inc. is located in Shiga prefecture, not Tokyo.

Answer:

As written in P.6, line 100-101, the trimmed aortas were cut along the longitudinal direction and were fabricated with an inner diameter of 15 mm with a hollow punch. Therefore, “the disked aorta” which is mentioned in Section 2.7 Cell seeding indicates 15 mm diameter punched aortas. We added the word “inner diameter 15 mm” in P.10, line 167.

As you have pointed out, we revised the location of TAKARA Bio Inc. in P.6, line 93. Thank you for your correction.

6) 2.9. qPCR

The authors forgot to indicate the Table of primers. There is no Table in the manuscript. I previously make an apology if I overlooked it.

Answer: 

We apologize for not indicating the Table of primers in the manuscript. We added the list of primers used in this experiment as Table 1 in P11, line 186. 

7) 2.10. Statistical analysis

As you know, the one-way ANOVA is suitable for evaluating the significant differences among more than three groups.

Answer:

We performed one-way ANOVA, Tukey’s multiple comparison test in Figure 3(F), 6(K), 7(K) and 8(A)(B). In Figure 3(F), the F-CHP intensity was compared among different groups. In Figure 6(K) and 7(K), the cell density of each sample was compared by day. We added the symbols in those figures and the meaning of symbols were written in the legend of each figure. For figure 8(A) and (B), the relative expression of each gene was compared by samples, but the difference was not statistically significant. According to the above statistical analysis, we revised the sentence in the section of “Statistical Analysis” in P.12, line 189-195.

8) 3. Results and Discussion, line 154

Please remove “Discussion” because you subsequently prepared “4. Discussion”.

Answer:

 Noted. We deleted the “Discussion” in P.12, line 197. Thank you for your correction.

9) 3. Results, line 210-211

“Glass and PTFE were used as negative and positive controls, respectively.”

Is glass a positive control, right? I think the position between Glass and PTFE should be placed opposite. Please add the further details (manufacture and name) of the Glass and PTFE you used.

Answer: 

For blood clotting test, glass and polytetrafluoroethylene (PTFE) were used as a negative and positive control, respectively, since the absence of blood clot is considered as positive. These materials are often used as reference materials in studies of blood coagulation (4), (5). The exposure of blood to glass surface which has negatively charged surface greatly shortens the clotting time, since contact of blood with glass surface initiate contact activation of the platelets and Factor XII. On the other hand, PTFE is widely used as a biomaterial for blood-contacting medical devices because of its properties, such as good biocompatibility, chemical stability and anti-cellular adhesiveness.

 We added the details (manufacture and name) of Glass and PTFE in section “Evaluation of thrombogenicity of decellularized aortas.” in P.9-10, line 150-151 and 156-157.

Reference

4) Sperling C, Fischer M, Maitz MF, Werner C. Blood coagulation on biomaterials requires the combination of distinct activation processes. Biomaterials. 2009;30(27):4447-56.

5) Li JM, Singh MJ, Nelson PR, Hendricks GM, Itani M, Rohrer MJ, et al. Immobilization of human thrombomodulin to expanded polytetrafluoroethylene. Journal of Surgical Research. 2002;105(2):200-8.

10) 3. Results, line 226 – 227

“It is widely known that endothelial cell seeding on luminal surface of cardiovascular grafts are required to prevent thrombus formation (9, 10).”

This sentence is complicated for me. What did you mean by “cell seeding”? Is “cell seeding” meaning that cells are seeded on cardiovascular grafts ex vivo?

Also, “,,,,,grafts are” should be corrected to “,,,,,,grafts is” in this sentence.

Answer: 

We apologize for confusing you. We revised above sentence as “It is widely known that endothelial cell coverage on luminal surface of cardiovascular grafts immediately after transplantation are required to prevent thrombus formation” in P.17, line 272-274. 

As you mentioned, we revised the “are” to “is” in P.17, line 273. Thank you for your correction.

11) Figure 3

Were the parameters of CLSM fixed for the observation of all samples? How did you adjust and calibrate the fluorescence intensity for all observations? The fluorescence intensity should be calibrated because the tissue has self-fluorescence and you can control the fluorescence intensity by changing the parameter for CLSM observation.

I recommend doing the quantitative evaluation of collagen denaturation using the fluorescence intensity of these pictures.

Answer:

We used fluorescence microscope (BZ-X710, Keyence Corp., Osaka, Japan)), not CLSM. The photos of Figure 3 were taken at same exposure time, 1/40 sec.　

According to your suggestions, F-CHP staining was analyzed in ImageJ software using measurements of the mean intensity and area of all remaining pixels after a background subtraction. The signal quantification was shown as integrated intensity (the mean intensity times area). We added this graph as Figure 3(F) in the manuscript. 

Figure. The integrated F-CHP signals quantifies from images of tissue section. Numbers are presented as mean + standard deviation. The * sign indicates statistical significance in comparison to the untreated aorta (p < 0.01). The ** sign indicates statistical significance in comparison to the HHP (intima) (p < 0.01). The *** sign indicates statistical significance in comparison to the HHP 90°C (intima) (p < 0.01).

12) Fig. 6

I agree that both types of cells recognized “something” on decellularized aortas and the “something” is relating to the denaturation of the collagen molecule. But, I cannot believe that cells directly recognize the denatured collagen because the amino acid sequence of collagen did not change by the denaturation process. The initial adhesion of HUVECs is similar between HHPi and HHPi-90 degrees, but the proliferation behavior of adherent HUVECs was completely different. This means that HUVECs recognize the adhesive sites in collagen molecule on both aortas as well. Then, HUVECs was affected by “something” and was not able to proliferate on HHPi-90 degrees. Do you have any speculation about “something”? Was the tissue structure of HHPi-90 degrees maintained during cell culture for 7 days?

Answer:

We consider that vascular cells seeded on each sample recognized the structure and component of 

luminal surface, and regulated adhesion, proliferation and functional expression. As for HHP intima and HHP 90°C intima, the number of adhered HUVECs was almost the same, but the morphology of them was different. HUVECs on HHP intima extended, while HUVECs on HHP 90°C intima didn’t extend and showed round shape. This phenomenon occurred on HHP 30min dried intima and tunica-media. Therefore, we assumed that HUVECs on HHP 90°C sloughed off during cell culture, so that the total number of HUVECs on day7 was few. We added above sentences in the manuscript in P.17, line 281-287. 

Also, the tissue structure of HHP90°C remained same during cell culture.

13) Fig. 8

Please add the enlarged view in Fig. 8 (A). It was so difficult to recognize a difference in the gene expression level of all samples. Please perform the statistical analysis. In my opinion, you can remove this figure if you do not use it for discussion.

Answer:

We added the enlarged view in Figure 8(A). The sentences about Fig 8(A) were mentioned in P.23, line 375-377.

14) 4. Discussion, line 314-315

“Cell seeding on decellularized aortas and the evaluation of mRNA expression of those cells by RT-PCR were performed.”

Does “those cells” mean both HUVECs and AOSMCs? If so, the gene expression analysis of HUVECs should be added to this manuscript.

Answer: 

We apologize again for confusing you. We did not perform the gene expression analysis of HUVECs. Therefore, we revised the sentence as “HUVEC and HAoSMCs were seeded on decellularized aortas, and the phenotype of HAoSMCs was evaluated by qRT-PCR.” in P.22-23, line 373-374.

15)

There are several incomprehensible sentences and grammatical errors in this manuscript. [For example, line 304-308 and 310-314]

I recommend that this manuscript should be checked carefully or get the English proofreading.

Answer:　

I requested native speakers of English to proofread our English writing after revising manuscript. I have attached the certificate of English editing.

Reviewer #2: The manuscript studies the effect of luminal surface structure on thrombus formation and cell behavior using HHP decellularized aortas. The luminal surface structure was disrupted by heating, drying and peeling. The resultant surfaces were analyzed by immunohistological staining, collagen hybridizing peptide (CHP) staining and scanning electron microscopy (SEM) analysis. Overall, the manuscript has a poor-written language and requires major revision in several parts. It lacks novelty, and the data is not interesting. Here are some points that may improve the quality of the manuscript:

Answer: We appreciate your valuable suggestions on our manuscript. We response the reviewer’s questions point by point as below.

1) The lines from 59 to 66 are copied from the abstract.

Answer:

We deleted the sentence which you mentioned.

2) Information (company-cat number for antibody) for chemicals reagents and instruments need to be mentioned in the materials and methods.

Answer:

We added detailed information of the chemical reagents in “Materials” in the Materials and Methods section in P.5-6, line 74-96. Detailed information of instruments was mentioned at each experiment method section.

3) How does the inner membrane peel off? explain more on methods.

Answer:

We apologize for not specifying the detailed method. To prepare tunica media, the outer membrane of the decellularized HHP aorta was held with tweezer while inner membrane was peeled off with another tweezer. We added the detailed methods in P.7, line 114-116.

4) Figures legends should not be in the manuscript.

Answer:　

We followed the PLOS ONE guidelines for figure caption which mentions “Each figure caption should appear directly after the paragraph in which they are first cited.” Please let us know if we misunderstood, we will revise them.

5) Why did the Author investigate the binding and proliferation of SMC in the lumen? These cells reside in medial layer.

Answer: 

 We would like to evaluate the effect of luminal surface structure of decellularized aorta on vascular cell attachment and proliferation. Therefore, we seeded endothelial cell and smooth muscle on the lumen of decellularized aorta and compared their behavior.

6) Scale bar in figure 2 is absent.

Answer:

We apologize for not mentioning the scale. Scale bar indicates 50μm. We added the scale in Figure 2 and also noted in figure legend.

7) CF-CHP staining images in Figure 3 have different contrast, limiting meaningful comparison between images.

Answer: 

We used fluorescence microscope (BZ-X710, Keyence Corp., Osaka, Japan)), and the photos of Figure 3 were taken at same exposure time, 1/40 sec.　

F-CHP staining was analyzed in ImageJ software using measurements of the mean intensity and area of all remaining pixels after a background subtraction. The signal quantification was shown as integrated intensity (the mean intensity times area). We added this graph as Figure 3(F) in the manuscript. 

Figure. The integrated F-CHP signals quantifies from images of tissue section. Numbers are presented as mean + standard deviation. The * sign indicates statistical significance in comparison to the untreated aorta (p < 0.01). The ** sign indicates statistical significance in comparison to the HHP (intima) (p < 0.01). The *** sign indicates statistical significance in comparison to the HHP 90°C (intima) (p < 0.01).

8) and 9) In line 197, the authors mentioned that SEM analysis showed thicker fibers of HHP 90 ºC tunica-intima. Quantification of fiber diameters from SEM images among the samples should be provided with statistical analysis.

 In line 199, from Fig. 4D and I, the authors indicated more gap between fibers on HHP 30 min dried tunica intima. The gap distance among the samples has to be quantified and analyzed statistically.

Answer: 

 As you mentioned, it has to be quantified and analyzed to assess the fiber thickness and gaps between fibers. We are very interested in your suggestion, but we consider that the difference won’t be statistically significant. Therefore, we revised the sentence in P. 15, line 242-247, not including fiber thickness and gaps.

10) In line 226, the paragraph has only one sentence.

Answer: 

 We inserted a line break by mistake. Thank you for your correction.

11) The images in figure 4G-I have different contrast compared with other images in the same figure.

Answer: 

 We adjusted the contrast of SEM images and changed Figure 4.

12) Statistical analysis in Figures 6K, 7K, and 8 is absent.

Answer: 

Statistical analysis was performed with one-way ANOVA, Tukey’s multiple comparison test in Figure 6(K), 7(K) and 8(A)(B). In Figure 6(K) and 7(K), the cell density of each sample was compared by day. We added the symbols in those figures. For figure 8(A) and (B), the relative expression of each gene was compared by samples, but the difference was not statistically significant.

13) The manuscript has grammatical errors, several too-short sentences, and the language needs revision.

Answer:　

I requested native speakers of English to proofread our English writing after revising manuscript. I have attached the certificate of English editing.

---

## [Decision Letter · Decision Letter 1]

4 May 2021

Effect of luminal surface structure of decellularized aorta on thrombus formation and cell behavior

PONE-D-21-01350R1

Dear Dr. Kishida,

We’re pleased to inform you that your manuscript has been judged scientifically suitable for publication and will be formally accepted for publication once it meets all outstanding technical requirements.

Kind regards,

Michiya Matsusaki

Academic Editor

PLOS ONE

Additional Editor Comments (optional):

Reviewers' comments:

Reviewer's Responses to Questions

**Comments to the Author**

1. If the authors have adequately addressed your comments raised in a previous round of review and you feel that this manuscript is now acceptable for publication, you may indicate that here to bypass the “Comments to the Author” section, enter your conflict of interest statement in the “Confidential to Editor” section, and submit your "Accept" recommendation.

Reviewer #1: All comments have been addressed

2. Is the manuscript technically sound, and do the data support the conclusions?

Reviewer #1: Yes

3. Has the statistical analysis been performed appropriately and rigorously? 

Reviewer #1: Yes

4. Have the authors made all data underlying the findings in their manuscript fully available?

Reviewer #1: Yes

5. Is the manuscript presented in an intelligible fashion and written in standard English?

Reviewer #1: Yes

6. Review Comments to the Author

Reviewer #1: Dear Authors,

(PONE-D-21-01350R1)

Thank you for your earnest responses to my comments. I have confirmed the revised manuscript is addressing all comments. I recommend accepting the revised manuscript.

7. PLOS authors have the option to publish the peer review history of their article (what does this mean?). If published, this will include your full peer review and any attached files.

Reviewer #1: No

---

## [Editor Report · Acceptance letter]

7 May 2021

PONE-D-21-01350R1 

Effect of luminal surface structure of decellularized aorta on thrombus formation and cell behavior 

Dear Dr. Kishida:

I'm pleased to inform you that your manuscript has been deemed suitable for publication in PLOS ONE. Congratulations! Your manuscript is now with our production department. 

Kind regards, 

on behalf of

Dr. Michiya Matsusaki 

Academic Editor

PLOS ONE